# Subcellular Targeting of Plant Sucrose Transporters Is Affected by Their Oligomeric State

**DOI:** 10.3390/plants9020158

**Published:** 2020-01-27

**Authors:** Varsha Garg, Aleksandra Hackel, Christina Kühn

**Affiliations:** Humboldt-Universität zu Berlin, Plant Physiology, Philippstr. 13, Building 12, 10115 Berlin, Germany; varsha.garg.1@hu-berlin.de (V.G.); hackelal@staff.hu-berlin.de (A.H.)

**Keywords:** subcellular targeting, vesicle traffic, membrane microdomains, sucrose transporter, endocytosis

## Abstract

Post-translational regulation of sucrose transporters represents one possibility to adapt transporter activity in a very short time frame. This can occur either via phosphorylation/dephosphorylation, oligomerization, protein–protein interactions, endocytosis/exocytosis, or degradation. It is also known that StSUT1 can change its compartmentalization at the plasma membrane and concentrate in membrane microdomains in response to changing redox conditions. A systematic screen for protein–protein-interactions of plant sucrose transporters revealed that the interactome of all three known sucrose transporters from the Solanaceous species *Solanum tuberosum* and *Solanum lycopersicum* represents a specific subset of interaction partners, suggesting different functions for the three different sucrose transporters. Here, we focus on factors that affect the subcellular distribution of the transporters. It was already known that sucrose transporters are able to form homo- as well as heterodimers. Here, we reveal the consequences of homo- and heterodimer formation and the fact that the responses of individual sucrose transporters will respond differently. Sucrose transporter SlSUT2 is mainly found in intracellular vesicles and several of its interaction partners are involved in vesicle traffic and subcellular targeting. The impact of interaction partners such as SNARE/VAMP proteins on the localization of SlSUT2 protein will be investigated, as well as the impact of inhibitors, excess of substrate, or divalent cations which are known to inhibit SUT1-mediated sucrose transport in yeast cells. Thereby we are able to identify factors regulating sucrose transporter activity via a change of their subcellular distribution.

## 1. Introduction

Sucrose transporters are essential proteins for partitioning of photoassimilates within higher plants and down-regulation of sucrose transporter expression often results in drastic physiological changes regarding plant growth, fertility, flowering behavior, biotic interactions and stress tolerance [1,2,3,4]. Therefore, regulation of sucrose transporters has extensively been studied at various levels [5,6], whereupon only recent studies focus on the post-translational control of sucrose transporters that can occur by phosphorylation, oligomerization, protein-protein-interaction or subcellular redistribution [7].

Subcellular redistribution of plasma membrane proteins is one possibility to regulate the permeability of a given membrane for the corresponding substrate at the post-translational level in a very short time frame and in a very tight manner. Therefore, endocytosis and exocytosis events represent an attractive tool to manipulate the activity of transporter proteins. We observed that not only homo- and heterodimerization of sucrose transporters affect their transport capacity, but also some of the protein–protein-interaction partners that have been identified by the split ubiquitin system in yeast cells and afterwards confirmed *in planta* by alternative methods [2,8].

We screened systematically using the solanaceous sucrose transporters StSUT1 and SlSUT2 as bait proteins and identified a couple of very interesting candidates which might affect subcellular targeting of the transporter such a SNARE proteins, SEC61 proteins, chaperones, etc. SlSUT2 was shown to interact with a v-SNARE /VAMP711 protein, with the auxin carrier AUX1 and with three different disulfide isomerases [2] Here, we aim at elucidation of the impact of SlSUT2-interacting proteins on its subcellular distribution.

StSUT1 was shown to be associated to lipid raft-like microdomains in a redox-dependent manner and recycled in highly motile vesicles at the sieve element plasma membrane [9]. In the presence of high sucrose concentration (>500 mM sucrose), StSUT1 endocytosis is stimulated [10], whereas at lower sucrose concentration most of the protein is detected at the plasma membrane. Similar observation was made in the presence of brefeldin A, cycloheximide, or both, enhancing vesicle formation and StSUT1 internalization [11].

If raft-association is disturbed by application of methyl-beta-cyclodextrin (MβCD), which depletes the plasma membrane of sterols, the protein is evenly distributed over the membrane and the effect of brefeldin A is diminished [9].

We therefore concluded that endocytosis of StSUT1 seems to be raft-dependent and association of StSUT1 to membrane microdomains a pre-requisite of its internalization [9].

Not only inhibitor treatments were able to impact sucrose transporter distribution within plant cells, but also post-translational modifications.

In a previous study, we already showed that the dimerization of StSUT1 *in planta* seems to be daytime dependent with more protein in its dimeric form in the middle of the light period (8 h after light onset in LD conditions) when sucrose export form leaves is maximal and sucrose content in leaves is highest [12]. In parallel, we were able to detect more of the dephosphorylated form at the same time, whereas StSUT1 migrates only as a single (most likely phosphorylated form) at the end of the night [12].

One assumption could be that dephosphorylation of the protein is required for homodimerization or *vice versa*, homodimer formation precedes dephosphorylation. Here, we will investigate the effect of homodimerization of different sucrose transporters on their subcellular targeting and function. Surprisingly, opposite effects can be observed depending on which sucrose transporter-like protein will be investigated.

In order to fill the knowledge gap regarding regulation of subcellular dynamics of sucrose transporters, we aim at elucidation of the impact of homodimer formation, interaction with protein–protein interaction with other factors such as secondary messengers like divalent cations of the subcellular distribution of sucrose transporters.

## 2. Results

### 2.1. Homodimerization Affects Subcellular Distribution of Sucrose Transporters in An Opposite Way

Bimolecular fluorescence complementation (BiFC) was used to confirm the ability of StSUT1 to form homodimers *in planta* [12]. The homodimeric form of StSUT1 was not only detectable at the plasma membrane, but also in intracellular structures of 0.5−1 µm diameter [12].

As already observed for the StSUT1 homodimer (Figure 1B, [12]), an increased number of intracellular vesicles can be observed for the sucrose transporter StSUT4 in BiFC experiments (Figure 1C,D). StSUT1–StSUT1 homodimers are considered as a positive control in BiFC experiments since homodimer formation was published earlier [12], whereas infiltration of the StSUT4-VYCE construct together with the viral p19 repressor was used as a negative control (Appendix A).

Regarding the localization of the monomer versus localization of the dimer, the opposite effect was observed for SlSUT2: SlSUT2-YFP is retained intracellularly in intracellular vesicles if expressed transiently in epidermis cells of leaves of *Nicotiana benthamiana* (Figure 2B), as well as in stably transformed plants of *Nicotiana tabacum* (Figure 2C; [2]). If SlSUT2-YFP is heterologously expressed in yeast cells, it is enclosed in vacuolar-like organelles, one possible reason for its non-functionality in yeast complementation assays [14]. In BiFC experiments however, the homodimer of SlSUT2 is observed mainly at the plasma membrane 4–5 d after infiltration (Figure 1E,F and Figure 2D). In case of homodimerization of SUT2, dimer formation obviously enhances plasma membrane targeting, whereas in case of SUT1 and SUT4 enhanced internalization of the homodimers can be observed (Figure 1).

Interestingly, a very similar phenomenon was observed for the orthologous sucrose transporter from Arabidopsis: AtSUT2-dsRED transiently expressed under control of the CaMV35S promoter in protoplasts of *Nicotiana benthamiana* show vesicle formation with a diameter of 3.8 µm (Appendix A) in contrast to the empty vector coding of dsRED alone (Appendix A). In BiFC experiments however, the ability to form homodimers could be confirmed revealing a more efficient plasma membrane targeting of the AtSUT2 homodimer (Appendix A). Please note, that in contrast to the SlSUT2 transporter, the orthologous transporter from Arabidopsis, AtSUT2, shows very weak sucrose uptake capacity with low affinity towards the substrate [15].

We aimed at generation of a functional SlSUT2 tandem construct by cloning the SlSUT2 cDNA as a translational fusion construct with a short peptide linker sequence in between in an appropriate yeast expression vector in order to test whether the tandem construct would be functional in sucrose uptake in yeast, but this attempt failed (data not shown).

### 2.2. Heterodimers between SUT1 and SUT2 Are Internalized As Well

Co-expression of members of the SUT1 clade together with members of the SUT3/SUC3 clade of sucrose transporters often reduces the SUT1-mediated high sucrose uptake capacity. This was first quantified in yeast: StSUT1 expressed under the *Adh1* promoter in the yeast expression vector 112A1NE and co-expressed with StSUT1 expressed under the strong *Pma1* Promoter in pDR196 mediates sucrose uptake with a V_max_ of 267.45 nmol sucrose min^−1^ 10^6^ cells^−1^. When StSUT1 in 112A1NE is co-expressed with SlSUT2 under the strong *Pma1* promoter in pDR196, the uptake is lowered more than 100 fold to only 2.48 nmol min^−1^ 10^6^ cells^−1^ [16].

In order to answer the question whether heterodimerization between StSUT1 and SlSUT2 affects subcellular targeting in a way that sucrose uptake capacity at the plasma membrane will be affected, we performed BiFC experiments where the two halves of YFP were translationally fused to the C-terminus of the transporters. It should be taken into account that post-translational modifications such as phosphorylation by specific protein kinases might be different in the two expression systems, i.e., yeast cells versus *Nicotiana benthamiana*.

Regardless of whether StSUT1 was fused to the C- or the N-terminus of YFP and co-expressed transiently in infiltrated leaves of *Nicotiana benthamiana*, the heterodimer formation with SUT2 increased the amount of intracellularly localized oligomeric complexes (Figure 3A,B) compared to the monomeric form of StSUT1 mainly found at the plasma membrane (Figure 1A). Thus, not only homodimer formation, but also heterodimer formation seems to increase StSUT1 internalization. It cannot be excluded that increased internalization of StSUT1 is the main reason for the loss of function of the transporter if co-expressed with SUT2.

### 2.3. SlSUT2 Targeting to the Plasma Membrane Is Also Affected by Other Protein–Protein Interaction Partners

Systematic screening of an expression library for SlSUT2-interacting proteins revealed interactions with other transporters that are internalized by endocytosis like AUX1 or various proton ATPases or with components of the vesicle trafficking machinery like a v-SNARE/VAMP711 protein, three different protein disulfide isomerases probably acting as chaperones, ubiquitin-like proteins, receptor-like kinases, G protein-coupled receptors, and with proteins involved in brassinosteroid synthesis (DIM1) or BR signaling (MSBP1, BAK1) [2]. Some of those are described to be internalized by endocytosis or involved in vesicle trafficking (v-SNARE, MSBP1). The question would be whether co-expression with the SNARE protein affects subcellular localization of SlSUT2.

It is described for plasma membrane intrinsic proteins (PIPs) or potassium channels that they physically interact with the SNARE SYP121 syntaxin which affects potassium channel trafficking and activity [17].

First, we aimed at confirmation of interaction between the sucrose transporter SlSUT2 and the full-length v-SNARE/vesicle-associated membrane protein 711 protein via bimolecular fluorescence complementation (BiFC; Figure 4). Whereas SlSUT2 fused to YFP or VAMP711 alone fused to YFP showed subcellular retention in some intracellular vesicles (Figure 4A,B), the heteromeric complex shown in BiFC experiments is partially targeted to the plasma membrane as well (Figure 4C).

In the case of the SNARE SYP121 syntaxin, the dominant-negative SYP121-Sp2 fragment is sufficient to decreased the delivery of PIP2;5 or potassium channels to the plasma membrane [17,18]. The potassium channel KAT1 is localized in plasma membrane microdomains of 0.5 µm in diameter und the delivery of KAT1 to the plasma membrane is efficiently suppressed by the Sp2 fragment of SNARE SYP121 protein [19].

In order to test the efficiency of truncated SNAREs, we generated a truncated soluble form of v-SNARE/VAMP711 by deleting the C-terminal transmembrane domain of it (Figure 5) and co-expressed the truncated version of VAMP711 together with the SlSUT2 sucrose transporter in protoplasts of transformants of *Nicotiana tabacum* stably expressing a SlSUT2-YFP fusion construct (Figure 5). Here again, as previously shown for the full length VAMP711 protein (Figure 4), the plasma membrane targeting of SlSUT2-YFP could be enhanced significantly: whereas the monomeric form of SlSUT2-YFP is again mainly localized to intracellular vesicles (Figure 5B,C), the heteromeric complex is mainly detectable at the plasma membrane in single scans as well as in maximum projections (Figure 5D,E), indicating that even the soluble form of VAMP711 without transmembrane spanning domain is able to enhance plasma membrane targeting of SlSUT2.

### 2.4. StSUT4 Targeting Is Affected by Divalent Cations Calcium

There is still a lot of controversial discussion regarding the physiological function and localization of SUT4 sucrose transporters. Most of them have been localized to the vacuolar membrane if fused to GFP, but are functional at the plasma membrane as sucrose-proton-co-transporters if heterologously expressed in yeast or in Xenopus oocytes [20,21]. Since all known SUT4 sucrose transporters act as proton-symporters at the plasma membrane, a role in sucrose efflux from the vacuole was postulated since sucrose uptake into the vacuole is most likely done in a proton antiport mechanism and recently assigned to the family of TST proteins, formerly known as TMT proteins [22].

Also, the physiological function of SUT4 proteins seems to be species-dependent and induces varying phenotypic modifications in the appropriate mutants and transformants from rice, Arabidopsis, potato, or poplar plants [3,23,24]. For the *Lotus japonicus* LjSUT4 a role during nodule development was postulated [25], whereas the rice transporter OsSUT4 seems to be involved in seed germination and pollen maturation in a temperature-dependent manner [26]. In a recent study, an important role of SUC4 at the tonoplast was postulated to be responsible for stomatal movement and sensitivity towards drought stress [27].

Although the StSUT4 protein is a functional sucrose transporter at the plasma membrane of yeast cells, the detailed co-localization studies together with the plasma membrane marker protein CBL1-OFP does not reveal a clear co-localization of StSUT4-YFP if transiently expressed in leaves of *Nicotiana benthamiana* (Figure 6A). However, co-expression with the vacuolar marker protein PTR2-YFP (Figure 6C) clearly shows overlapping areas with the localization of StSUT4-RFP (Figure 6B,D). This is in agreement with other reports [20].

As revealed for all other sucrose transporters, the localization of SUT4 proteins seems to by dynamic and not restricted to one single subcellular compartment. Here, we provide further evidence for the subcellular mobility of a member of the SUT4 subfamily of sucrose transporters. As previously observed for the StSUT1 protein, increased internalization of the sucrose transporter can be induced by treatment with the translational inhibitor cycloheximide (10 µM, Figure 7A). Also, in case of treatment with 50 mM CaCl_2_ the labeling of the tonoplast diminished (Figure 7B), whereas the water treated control (Figure 7D) or the treatment with 50 mM EDTA (Figure 7C) still shows labeling of the tonoplast. It is obvious that subcellular localization of the StSUT4 protein seems to respond to divalent cations such as Ca^2+^.

## 3. Discussion

### 3.1. Dynamic Localization of Sucrose Transporters

Here we show that subcellular localization of sucrose transporters in transiently transformed leaves of *Nicotiana benthamiana* is dynamic and not restricted to one single compartment depending on environmental conditions, homodimer formation, protein–protein-interaction, calcium concentration, etc.

Sucrose transporters SlSUT2 and StSUT4 are expressed at very low levels *in planta* and the usage of a strong and constitutive promoter, such as the CaMV35S promoter which was used in this study, is a powerful tool allowing visualization of even low abundant proteins. Nevertheless, overexpression in heterologous systems under control of foreign promoters might harbor a source of artifacts and care should be taken while interpreting the results.

In our studies, StSUT1 and StSUT4 behave very similar, whereas SlSUT2 which never showed sucrose uptake capacity in heterologous systems rather showed the opposite behavior. Homodimerization increased internalization of StSUT1 and StSUT4 as well as of the heterodimers with SUT2, whereas homodimers of SlSUT2 or heteromeric complexes with the v-SNARE/VAMP711 or the truncated versions of it increased plasma membrane targeting of SlSUT2.

Most likely, co-expression of SlSUT2 inhibited StSUT1-mediated sucrose uptake by increasing the internalization of the heterodimeric complex.

Interestingly, very similar results are obtained for homodimers of AtSUT2 (Appendix A), a sucrose transporter which is also shown to undergo homo- as well as heterodimerization [28].

### 3.2. Putative Effect of Sucrose Transporter Phosphorylation

A double band detectable on SDS-PAGE for StSUT1 protein most likely represents the phosphorylated and the de-phosphorylated form with increasing amounts of the dephosphorylated form of StSUT1 at the end of the light period [12]. In parallel, homodimerization occurs simultaneously with increasing amount of the homodimer detectable under non-reducing conditions and homodimerization accompanies dephosphorylation and increase in transport activity which is maximal at the end of the light period [12].

This would fit with observations in *Beta vulgaris* where BvSUT1 is more active in its dephosphorylated form as shown by inhibitor studies [5,6]. Okadaic acid affects BvSUT1 sucrose transport activity by maintaining the transporter in its dephosphorylated form [5].

Now it is the question whether dephosphorylation of the protein is required for homodimerization or *vice versa* and whether dephosphorylation/homodimerization of the protein affects subcellular targeting.

A more efficient targeting to the plasma membrane would easily explain the gain in transport activity which is observed at the end of the light period *in planta*, when sucrose export form source leaves is highest.

Cis-regulatory circadian promoter elements are responsible for the circadian expression of sucrose transporters with maximum transcript amounts for StSUT1 and StSUT4 in the middle of the light period [3]. Regarding transient expression of sucrose transporter in *N. benthamiana* under control of the strong and constitutive CaMV35S promoter, a circadian regulation of the transcript amount is not expected.

### 3.3. Effect of Divalent Cations on Sucrose Transporter Activity and Localization

In this context, it is interesting to note that the activity of the StSUT1 transporter responds to pre-treatments with calcium or magnesium ions. Both divalent cations had inhibiting effects on sucrose transport activity of StSUT1 in ^14^C-sucrose uptake experiments in yeast [12]. It is the question whether this inhibition of sucrose uptake is due to increased internalization of StSUT1 by divalent cations. Further experiments are needed for this.

Interestingly, a highly conserved diacidic DTD motif in the seventh transmembrane spanning domain of StSUT1 seems to be involved in this inhibitory effect: if the DTD motif is mutagenized to GTG, the inhibitory effect of calcium ions is diminished [12].

It is worth mentioning that the activity of the *Solanum tuberosum* sucrose synthase 1 can be stimulated by Ca^2+^ and Mg^2+^, whereas Cu^2+^, Ni^2+^, and Zn^2+^ inhibit sucrose cleavage by sucrose synthase activity [29]. A similar phenomenon was also observed in *Oryza sativa* sucrose synthase activities [30]. In these cases, the involvement of calcium-dependent protein kinases is assumed [31]. In case of SUS1 form maize, it is described that phosphorylation of Serine15 at its N-terminus stimulates the sucrose cleavage activity of the enzyme [32], a process where membrane association of the enzyme plays a role.

It is still the question whether (calcium-dependent?) phosphorylation of sucrose transporters affect dimerization, subcellular localization, and thereby activity.

## 4. Methods

### 4.1. Recombinant DNA

All sucrose transporter and interaction partner cDNAs were cloned by GATEWAY technology after a two-step PCR protocol into the ENTRY vector pDONR221 and subsequently cloned via LR reaction in the appropriate destination vector as described previously [2,10,12,33]. For Bimolecular Fluorescence Complementation (BiFC) assay, full length sucrose transporter cDNAs were fused to N- and C-terminal sub-fragments of the yellow fluorescent protein (YFP) in the GATEWAY cloning vectors: pDEST-VYNE and pDEST-VYCE [13] to study homodimerization. For sub-cellular localization, full length StSUT4 cDNA was cloned in two destination vectors with different fluorescence marker protein; pK7YWG2.0 (YFP) and pH7RWG2.0 (RFP). The plasma membrane marker protein CBL1 is fused to OFP and Vacuolar marker protein PTR2 is fused to YFP. 

Primers used for GATEWAY cloning: StSUT1 fw: AA AAA GCA GGC TTA AAA ATG GAG AAT GGT ACA AAA AG; StSUT1 rev: A GAA AGC TGG GTA ATG GAA ACC GCC CAT GGC GAC; SlSUT2 fw: AA AAA GCA GGC TTA AAA ATG GAT GCG GTA TCG ATC; SlSUT2 rev: A GAA AGC TGG GTA ACC AAA ATG GAA GCC AGT TG; StSUT4 fw: AA AAA GCA GGC TTA ATG CCG GAG ATA GAA AGG CAT AG; StSUT4 rev: A GAA AGC TGG GTT TGC AAA GAT CTT GGG TTT CTC; SNARE fw: AA AAA GCA GGC TTA GAG ATG ACG ATA CTG TAT GCG C; SNARE rev: A GAA AGC TGG GTA ACG ATG AAA AAT TGC AAT ATA AT.

For the deletion of the 195 N-terminal nucleotides (deletion of 65 N-terminal amino acids including the transmembrane domain) of the v-SNARE/VAMP711 protein following primers have been used: Delta-SNARE fw: AAA AAG CAG GCT TAA TGA CGA TAC TGT ATG CG; Delta-SNARE rev: AGA AAG CTGBGGT TGC TTC TGA AAC GGC GAG C.

### 4.2. Transient Expression of Fluorescent Proteins

Transient expression in leaves of *Nicotiana benthamiana* was performed using *Agrobacterium tumefaciens* strain pGV2260 as described previously [9]. Viral P19 protein was used as a suppressor of post-transcriptional gene silencing. For agro-infiltration, the different combinations were mixed in 1:1 each with OD_600_ = 1.0. For BiFC experiments, the viral p19 repressor protein was co-infiltrated [34]. StSUT4 was cloned in the two destination vectors with different fluorescence marker protein; pK7YWG2.0 (YFP) and pH7RWG2.0 (RFP) and *Agrobacterium* strain GV2260 was used for transformation. The plasma membrane marker (PM) protein is CBL1-OFP and the vacuolar marker protein is PTR2-YFP; both were used for co-infiltration of *Nicotiana benthamiana*. Three- to four-week-old *Nicotiana benthamiana* plants were infiltrated at the lower epidermal side of the young leaves with the help of needleless syringe by applying equal pressure from upper side through finger. Three plants with four leaves each were infiltrated for one combination. The agro-infiltrated plants were kept at 22 °C, 16 h light/8 h dark cycle for 3−5 d in the greenhouse. The plants were observed under the Confocal microscope (Zeiss LSM 800 with Airyscan) after 3 d until 5 d by selecting different filters; Tag YFP, Tag OFP, and Tag RFP. To study the effect of different chemical treatments on the sub-cellular localization of StSUT4, the infiltrated leaves were dissected and incubated overnight at room temperature in different substrate solutions; 10 µM cycloheximide, 50 mM CaCl_2_, 50 mM EDTA, and 500 mM Sucrose by taking water as a control. The treated leaf discs were then transferred to water and observed under the Confocal Microscope (Zeiss LSM 800) by selecting Tag YFP and Tag OFP filters. 

## Figures and Tables

**Figure 1 plants-09-00158-f001:**
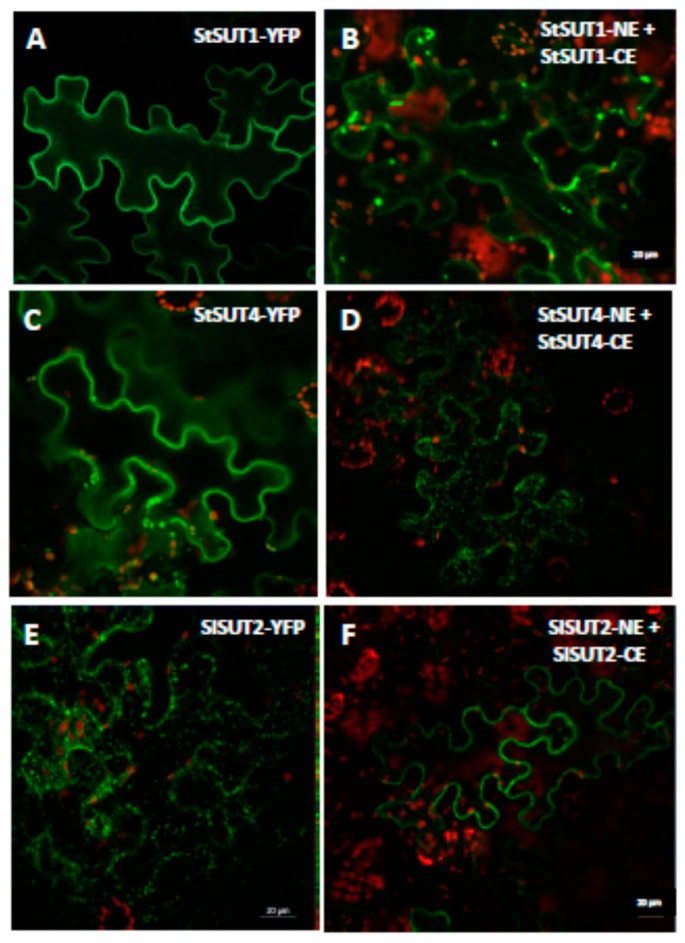
Homodimerization of sucrose transporters affects subcellular targeting. (**A**). StSUT1 in pK7GW2.0 (YFP) transiently expressed in *N. benthamiana* leaves. (**B**). BiFC experiment showing StSUT1 homodimerization were used as a positive control. (**C**). StSUT4 in pK7GW2.0 (YFP). (**D**). BiFC experiment showing StSUT4 homodimerization. (**E**). SlSUT2-YFP expression in *N. benthamiana* leaves. (**F**). BiFC experiment showing SlSUT2 homodimerization. For BiFC experiments sucrose transporter cDNAs were cloned in VYNE and VYCE vectors [13] and co-infiltrated together with the viral p19 repressor. All pictures were taken 4 d after infiltration.

**Figure 2 plants-09-00158-f002:**
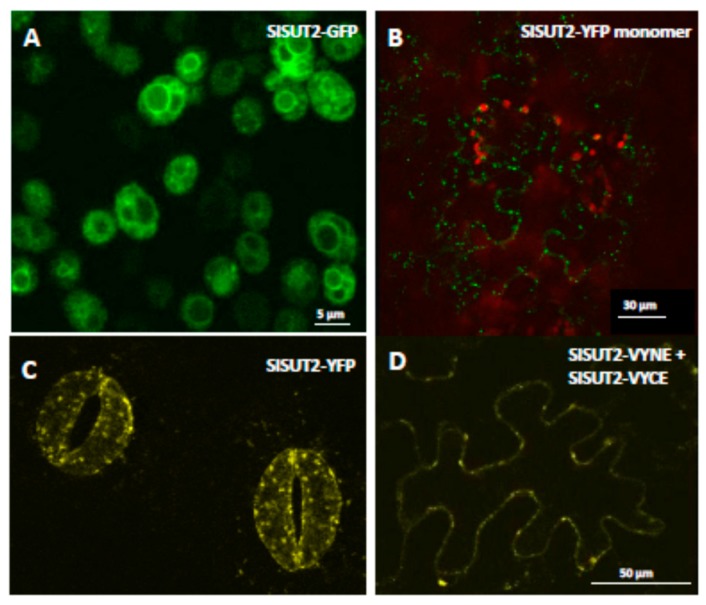
SlSUT2-GFP and SlSUT2-YFP fusion proteins heterologously expressed in yeast and in planta. (**A**). SlSUT2-GFP fusion construct expressed in *Saccharomyces cerevisiae* is not targeted to the plasma membrane. (**B**). SlSUT2 in pK7GW2.0 (YFP) transiently expressed in *N. benthamiana* leaves 4 d after infiltration. (**C**). Stable expression of SlSUT2-YFP under control of the CaMV35S promoter in *Nicotiana tabacum*. (**D**). BiFC experiments showing SlSUT2 homodimerization. Note that homodimers show increased plasma membrane targeting compared to the monomeric form of SlSUT2.

**Figure 3 plants-09-00158-f003:**
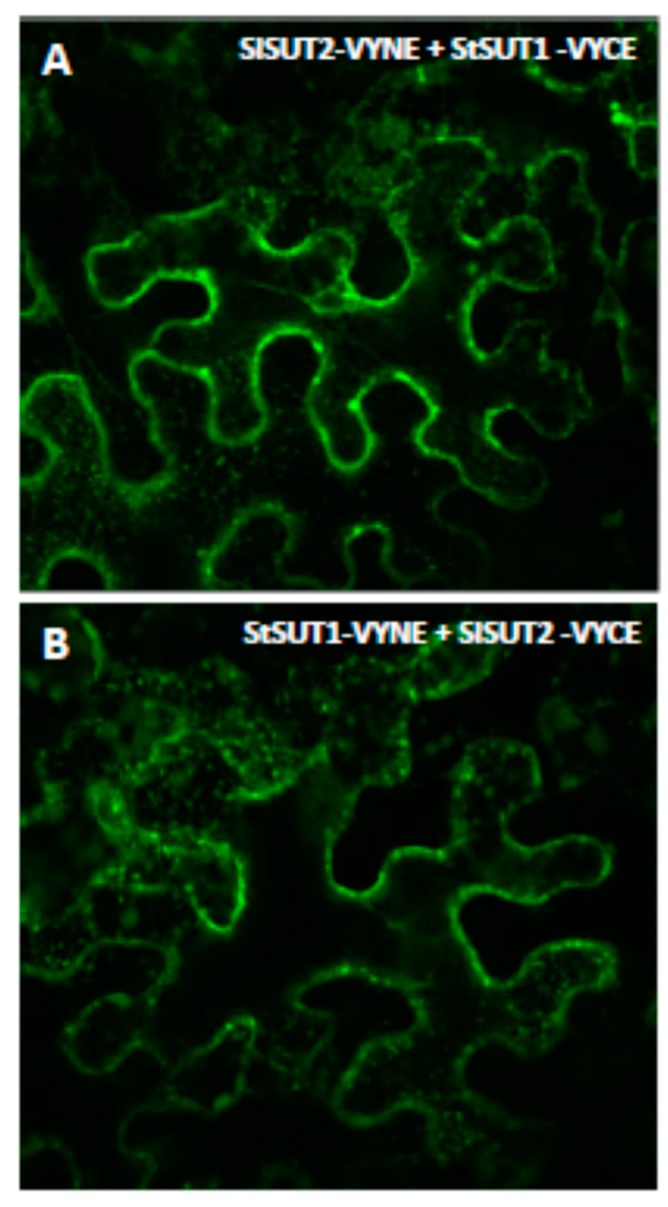
StSUT1 and SlSUT2 are able to form homodimers in planta (**A**). and (**B**). Confirmation of StSUT1-SlSUT2 heteromerization in planta by BiFC experiments using the vectors VYNE and VYCE (Gehl et al., 2009) in both possible combinations. Note that heterodimerization of StSUT1 and SlSUT2 increases internalization of the heteromeric complex.

**Figure 4 plants-09-00158-f004:**
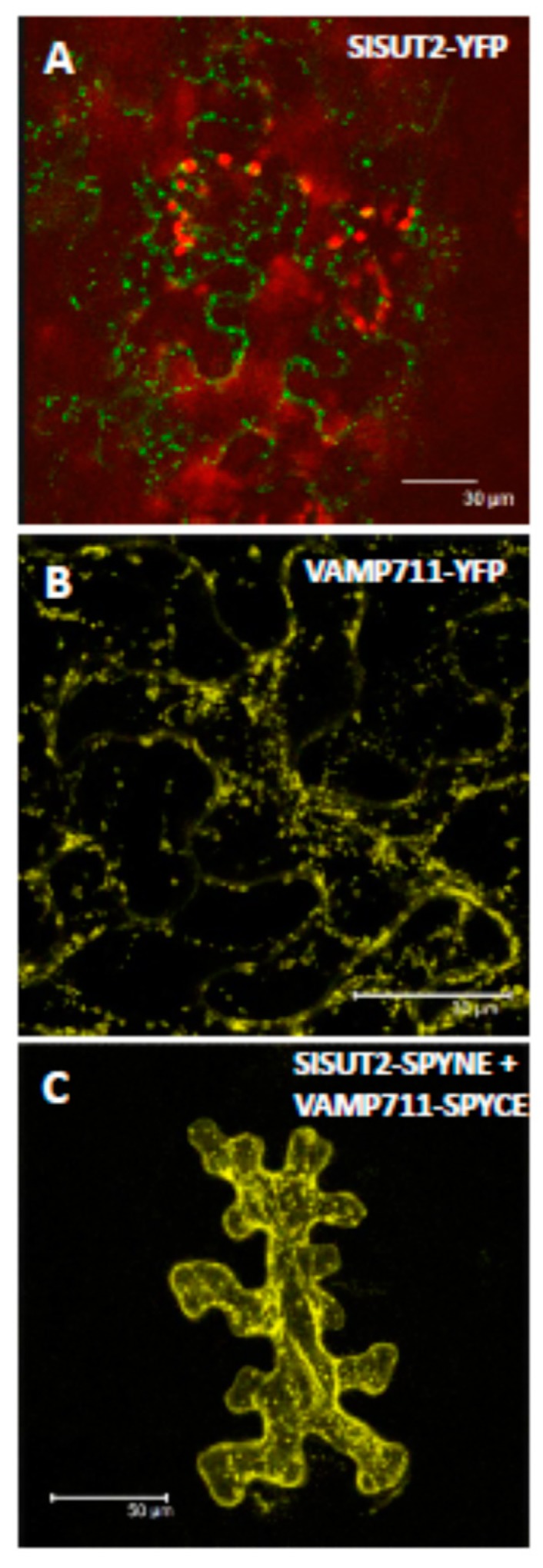
Confirmation of interaction between SlSUT2 and its interaction partner v-SNARE/VAMP711. (**A**). SlSUT2-YFP transiently expressed in leaves of *N. benthamiana* 4 d after infiltration. Chlorophyll is shown in red, YFP fluorescence is shown in green. (**B**). The full-length v-SNARE/VAMP711 in pK7GW2.0 (YFP) transiently expressed in leaves of *N. benthamiana*. YFP fluorescence is shown in yellow. (**C**). BiFC experiments confirming SlSUT2 and v-SNARE/VAMP711 interaction using the BiFC vectors SPYNE and SPYCE. Maximum projections of z-stacks are shown.

**Figure 5 plants-09-00158-f005:**
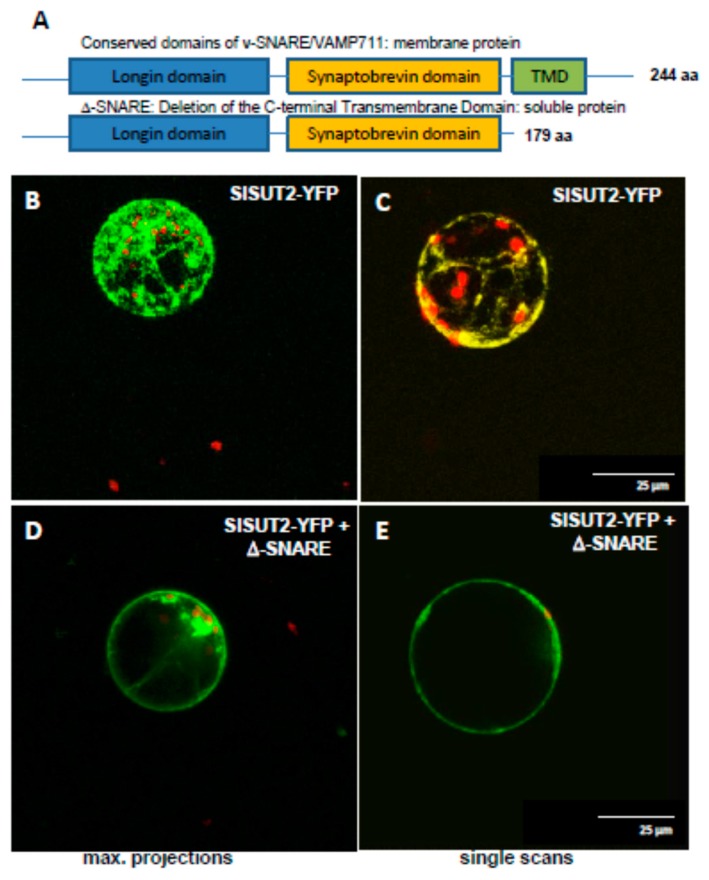
SlSUT2 co-expressed with the interacting VAMP711 protein. (**A**). Schematic representation of conserved domains within the v-SNARE/VAMP711 protein being one of the SlSUT2 interaction partners [2]. Upper panel: full length version of the protein; lower panel: D-SNARE protein with truncation of the C-terminal transmembrane spanning domain. (**B**). and (**C**). Protoplasts of *Nicotiana tabacum* transformed with a SlSUT2-YFP fusion construct. (**D**). and (**E**). Protoplasts co-expressing SlSUT2-YFP together with the truncated version of the v_SNARE/VAMP711 protein D-SNARE. Only the SlSUT2 protein is visible in green, whereas the D-SNARE protein is not tagged. B and D. Maximum projections of z-stacks, C and E. Single scans. Chlorophyll is shown in red, YFP is shown in green (**B**,**D**,**E**) or yellow (**C**).

**Figure 6 plants-09-00158-f006:**
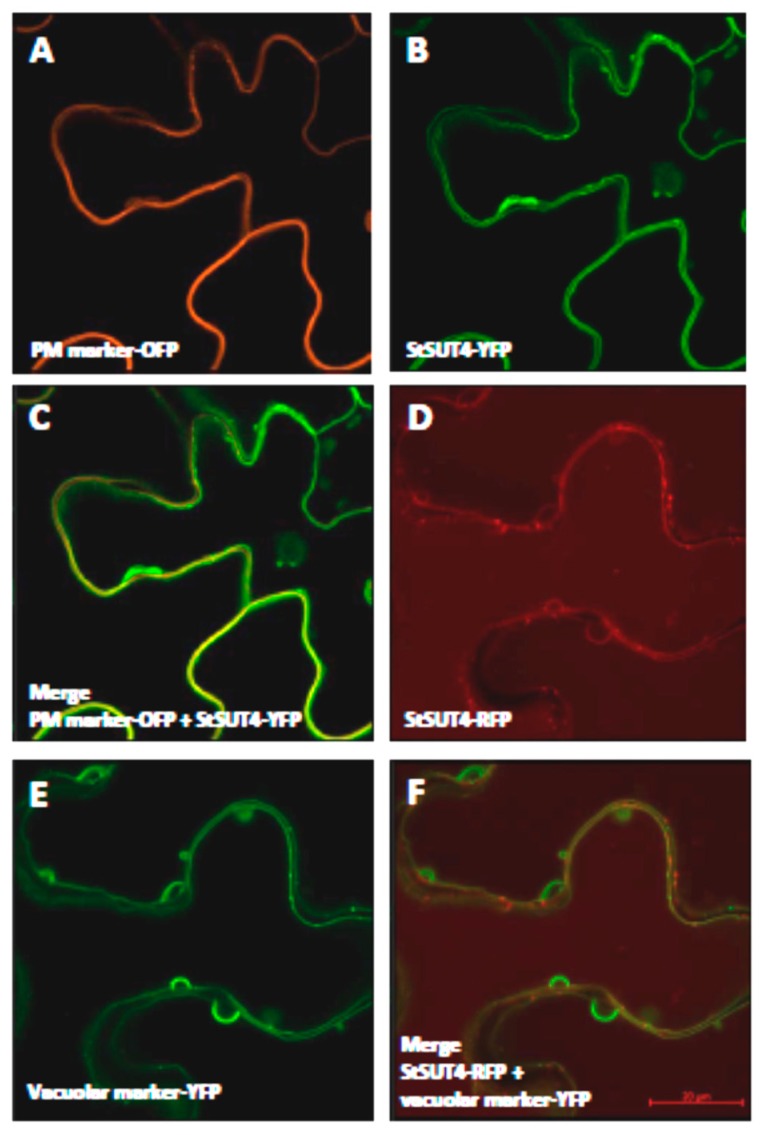
Colocalization of StSUT4 with different marker proteins. (**A**). Plasma membrane marker protein CBL1-OFP. (**B**). StSUT4-YFP alone. (**C**). Overlay picture of StSUT4-YFP and the PM marker CBL1-OFP. No co-localization of StSUT4 with the PM marker was observed. (**D**). StSUT4-RFP alone. (**E**). The vacuolar marker protein PTR2-YFP alone. (**F**). Overlay picture of StSUT4-RFP and PTR2-YFP. StSUT4 was co-localized with the vacuolar marker protein PTR2-YFP. Pictures were taken 4 d after infiltration. YFP fluorescence is given in green, OFP is given in orange, RFP is given in red.

**Figure 7 plants-09-00158-f007:**
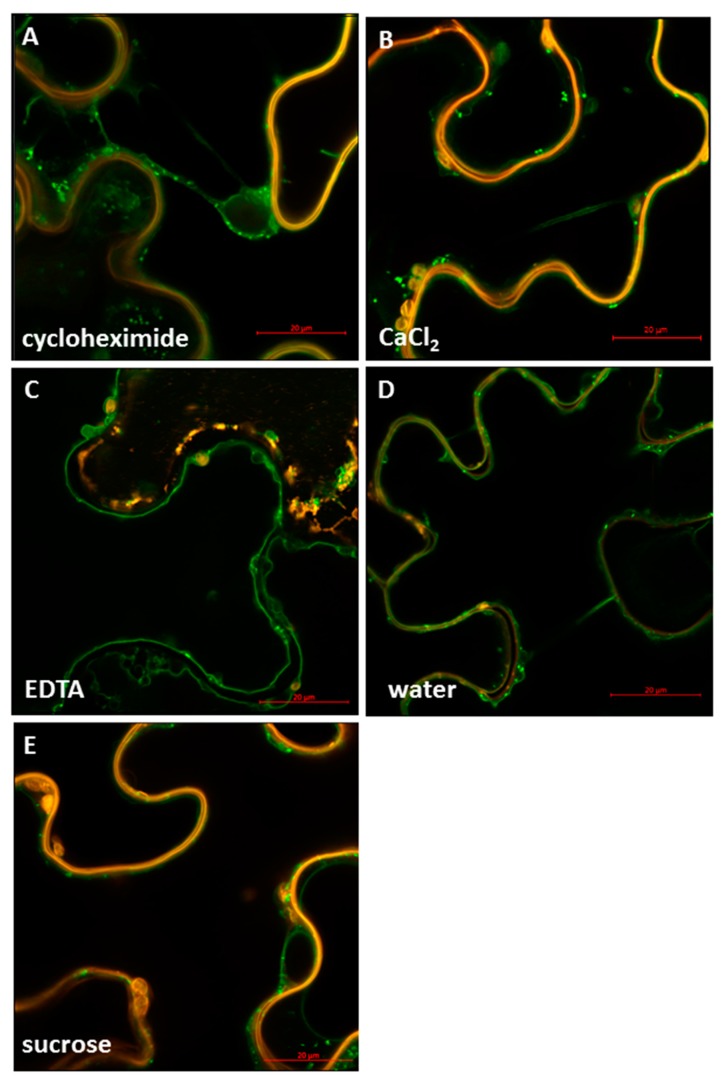
Effect of different effectors on the sub-cellular localization of StSUT4. The infiltrated leaves were incubated overnight at room temperature in different substrate solutions: 10 µM Cycloheximide (**A**), 50 mM CaCl_2_ (**B**), 50 mM EDTA (**C**), and 500 mM sucrose (**E**) using water as a control (**D**). The treated leaf discs were then transferred to water and observed under the Confocal Microscope (**A**). Movement of StSUT4-YFP towards perinuclear membranes and increase in the number of vesicles with no change of the PM marker (CBL1-OFP) after overnight incubation in 10 µM Cycloheximide. (**B**). Formation of vesicles in the presence of divalent calcium ions. (**C**). Overnight incubation in 50 mM EDTA affects the PM marker CBL1-OFP but not the localization of StSUT4. (**D**). No change in the localization of StSUT4 in the water control. (**E**). Movement of StSUT4-YFP in smaller vesicles in the presence of 500 mM sucrose.

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
