# Peer review of "Subcellular Targeting of Plant Sucrose Transporters Is Affected by Their Oligomeric State"

_plants, 2020, doi:10.3390/plants9020158_

Round 1

Reviewer 1 Report

In the manuscript, the factors regulating sucrose transporter activity via a change of their subcellular distribution were identified. Using BiFC experiments the authors demonstrated the role of homo and heterodimerization of SlSUT2 in membrane targeting and the role of divalent cations calcium in StSUT4 membrane targeting.

However, the manuscript should be improved. It should be carefully reread and revised in relation to misprints. In addition, figures should be presented in a more accurate way. I suggest that each Figure caption contains one general Caption and only then the description of A, B, C, D… sections follows (relevant to Figures 5 and 6). Please, carefully check if all Latin names and gene names are in italics.

Line 80 “As already observed 80 for the StSUT1 homodimer (Fig. 1B, [3], an increased number”

change to

“As already observed 80 for the StSUT1 homodimer (Fig. 1B, [3]), an increased number”

Line 146 “diameter und the delivery of KAT1 to the plasma”

change to

“diameter and the delivery of KAT1 to the plasma”

Line 182 “the localization of SUT4 proteins seems to by”

change to

“the localization of SUT4 proteins seems to be”

Lines 184-185 “As oreviously observed for the StSUT1 protein”

change to

“As previously observed for the StSUT1 protein”

Line 209 “form of StSUT1 at the end of the light period [3] Fig. 1E.”

I cannot understand if a reference to Fig. 1E is relevant in this sentence.

Line 170, 321-322 Lotus japonicus – in italics.

Lines 313, 317 Populus  - in italics

Line 325  OsSUT4 - in italics

Line 330 Arabidopsis - in italics

Lines 51-53, 56-57, 73-74, 219-221, and References –the font should be uniform according to the requirements of the journal.

Fig.1 please, make the size of the figures uniform, the white frames between figures uniform and more accurate, make the scale bars more visible (where they are present) and add them on other figures (where they are absent, Figure 1A,B,C,D; Figure 2A,C; Figure 3A,B; Figure 5B,D; Figure 6B,C,D; Figure 7 (all))

Figure 2. Please, change the location and font of the captions (such as Nocotiana tabacum, SlSUT2 variants etc.) In a present form it is very hard to understand since the direction and the font size of the text is different. Make it more accurate and uniform, organize captions in more uniform and readable way.

Figure 3 Remove “3” from the left top corner of the figure.

Figure 4 Remove “Figure 4” from the left top corner of the figure.

Figure 5 Letter A in the left top corner is partially cut. Correct it.

Please, make all Figure captions as a separate text not as an image.

The manuscript provides new and interesting results and can be published after corrections.

Kind regards.

Author Response

We would like to thank you and the reviewers for the helpful comments. All criticisms are certainly justified and helped us to improve the quality of our manuscript.

Responses to reviewer #1:

All figures have been improved and are now presented in a more accurate way. Each figure now contains a general caption. All latin names are in italics Line 80: brackets are inserted Line 146 “und” was replaced by “and” Line 182: “seems to by” is now corrected Line 184 : “oreviously” is now corrected Line 209: the text refers to Fig. 1E from reference Krügel et al. 2013. This is now corrected. Line 170 and others: all latin names are now in italics and the references have been reformatted. 1: figures were resized and uniform. The scale bars are now better visible and new scale bars were introduced. 2: all fonts and captions have been changed as suggested by the reviewer. 3, 4, 5: were corrected as suggested All figure captions are provided as a separate file.

Reviewer 2 Report

Subcellular targeting of plant sucrose transporters is affected by their oligomeric state

In this paper, Varsha Garg et al., have identified regulation of sucrose transporter localization due to either protein-protein interaction or due to change in cellular conditions. The authors have used the BIFC to assay homo and heterodimerization of StSUT transporters and subsequent localization of these proteins which is also dependent on spatio-temporal co-expression of these transporters. Also they confirmed the interaction with other protein complexes such as VAMP/SNARE and how it influences localization of SUT. In addition the effect of divalent cations and its influence on SUT localization has been shown. The results from this is crucial to understand the function of sucrose transporter and how its localization can be altered by multiple factors.

Major comments:

The materials and methods section is incomplete to interpret the results. The information about cloning of BIFC vector, microscopy settings, image acquisition, image analysis and any changes to figures, chemical treatment (Ca, EDTA) whether in dissected tissue or via injection in to infected area? The paper mainly discusses about subcellular localization, yet it is difficult to interpret whether the localization is in plasma membrane vs vacuole apart from vesicle formation. In Fig 1B, 1C, 1D, 1F, 2B, 2D and 3. Please show co expression of marker for PM and vacuole to interpret the localization data accurately. This will be crucial to understand the transport activity of the SUT.

Minor Comments:

Some minor suggestions on the manuscript: The flow of text is introduction could be improved and it could be made better with a) implying the significance of this study and b) how this is crucial inplanta for growth and development. C) Then state the regulation of these transporter family and the knowledge gap. D) Finally the hypothesis to be tested that might answer the knowledge gap.

Line 35: change realized to “observed” or “determined”.

Line 35-38: include reference.

Line 41: include reference.

Line 43: it will be easier as a reader to learn about known facts and knowledge gap in the beginning of introduction and the hypothesis with main outcome in the last paragraph.

Line47: Please check the reference format

Line 51: Please expand the abbreviation of gene names, chemicals when introducing for the first time in the text.

Fig 1 and following figures: Please add a positive and negative control for BIFC experiments to interpret the results accurately.

Fig 2A. Show an enlarged image of a yeast cell showing SlSUT2 expression.

-In the discussion , please mention about the strength and weakness of interpreting the results from overexpression in tobacco epidermal tissue.

-While comparing about localization/transport activity of SUT2 and SUT2 with other SUT transporter with earlier publication (Reinders A et al., 2002) please iterate about the yeast vs tobacco system. Is there a possibility of other regulators and microdomain protein complex influencing the results.

- Also discuss about how this localization knowledge can be useful for SUT orthologs of other plant species where homo and heterodimerization is already known.

Line 58 -61 suggest dimerization state varies depending on the clock. Does the results from this paper have any impact based on when the images were acquired?

Finally the paragraphs in introduction needs formatting and similarly the references. 

Author Response

Responses to reviewer #2:

Major comments:

The material and method section has been completed with additional paragraphs and a complete list of primers used for GATEWAY cloning of cDNAs. All details about incubation with chemical effectors are given. Regarding subcellular localisation of the StSUT4 protein, the figure 6 was rearranged with more co-localization images including vacuolar and plasma membrane markers. All airyscan images shown in figure 7 are performed in the presence of the plasma membrane marker CBL1-OFP. Regarding figures 1 and 2 they are not at high resolution (no Airyscan images as shown in Figure 6 and 7). Therefore the discrimination between plasma membrane and vacuole is difficult. If we need to repeat all experiments shown in Fig. 1, 2 and 3 und high resolution conditions, we need more time than 5 days for the revision of the manuscript since Agrobacterium-mediated infiltration of Nicotiana leaves will need preparation, plant growth and detailed evaluation by confocal microscopy.

Minor comments:

The introduction of the manuscript has been changed according to the suggestions of the reviewer. Additional paragraphs have been added and the existing ones have been improved. Line 35: “realized” has been replaced Line 35-38, 41: references have been included Line 43: the introduction has been changed accordingly Line 47: the reference has been reformatted Line 51: gene names and chemicals are now explained 1: the positive control of BiFC experiments is the homodimer formation of StSUT1 since this was originally published by Krügel et al. 2013. A negative control is given in the new Supplementary Figure S2. 2A: figure of SlSUT2 expression in yeast is now enlarged. In the discussion the advantages and disadvantages of transient expression in Nicotiana leaves are mentioned. The differences between heterologous expression in yeast and tobacco are discussed now. The only plant species where dimerization of sucrose transporters is already known to my knowledge is Arabidopsis (Schulze et al. 2003). This is now mentioned in the discussion. Line 58-61: an additional paragraph about circadian regulation of sucrose transporter expression as added. Paragraphs of the introduction and the references have been reformatted.